# Supplementary Materials for
# TRACE: A Comprehensive Benchmark for Continual Learning in Large Language Models

## I. DETAILS FOR DATASET DISTRIBUTION

TRACE is a benchmark designed for comprehensive continual learning evaluation for LLMs. You can download it from `https://drive.google.com/file/d/1S0SmU0WEw5okW_XvP2Ns0URflNzZq6sV/view`. It contains 8 datasets for evaluating a variety of capabilities of LLMs, each of which contains 500/1,000/5,000 instances for training and 2,000 instances for evaluation sampled from the original dataset. The datasets included in our benchmark are ScienceQA, FOMC, MeetingBank, C-STANCE, 20Minuten, the line-level code completion task of CodeXGLUE and the first two tasks of NumGLUE. We select these datasets because they are not only novel enough that most LLMs have not been trained on them, but also challenging for large language models. The overview of dataset statistics is included in the appedix of the paper.

## II. DATASHEET FOR DATASETS

**This document is based on *Datasheets for Datasets* by Gebru *et al.* [1]. Please see the most updated version here.**

### MOTIVATION

**For what purpose was the dataset created?** Was there a specific task in mind? Was there a specific gap that needed to be filled? Please provide a description.
The TRACE benchmark was designed to address significant gaps in evaluating the continual learning capabilities of large language models (LLMs), specifically aligned LLMs. Previous benchmarks were inadequate due to their simplicity and homogeneity, failing to challenge the complexities of current models. TRACE was created to provide a rigorous, multifaceted benchmark that includes tasks in domain-specific knowledge, multilingual capabilities, code generation, and mathematical reasoning, aimed at testing and improving the resistance of LLMs to catastrophic forgetting while ensuring they adapt efficiently to new tasks without losing prior knowledge.

**Who created this dataset (e.g., which team, research group) and on behalf of which entity (e.g., company, institution, organization)?**
TRACE was developed by an anonymous research group, aiming to contribute to the broader academic community's understanding of continual learning in AI. The development team's affiliation is not specified in detail in the paper provided.

**What support was needed to make this dataset?** (e.g.who funded the creation of the dataset? If there is an associated grant, provide the name of the grantor and the grant name and number, or if it was supported by a company or government agency, give those details.)
The paper does not specify any particular funding sources or grants for the creation of the TRACE dataset. It's common for such projects to be part of funded research initiatives by academic institutions or through collaborations that might not be directly funded by a specific grant.

**Any other comments?**
No additional comments were specified regarding the creation of the dataset.

### COMPOSITION

**What do the instances that comprise the dataset represent (e.g., documents, photos, people, countries)?** Are there multiple types of instances (e.g., movies, users, and ratings; people and interactions between them; nodes and edges)? Please provide a description.
The dataset mainly compromise text data, including documents, review, conference abstract, math problems, etc. Our benchmark consists of eight distinct datasets including domain-specific tasks, multilingual capabilities, code generation, and mathematical reasoning.

**How many instances are there in total (of each type, if appropriate)?**
As these datasets have varying sizes, we create a balanced version by randomly sampling 5000 training examples and 2000 testing examples from the original datasets. We get 40,000 training examples and 16,000 testing examples in total.

**Does the dataset contain all possible instances or is it a sample (not necessarily random) of instances from a larger set?** If the dataset is a sample, then what is the larger set? Is the sample representative of the larger set (e.g., geographic coverage)? If so, please describe how this representativeness was validated/verified. If it is not representative of the larger set, please describe why not

(e.g., to cover a more diverse range of instances, because instances were withheld or unavailable).

As these datasets have varying sizes, we create a balanced version by randomly sampling 5000 training examples and 2000 testing examples from the original datasets. If the original data set has less than 5000 entries, we oversample; otherwise, truncate.

**What data does each instance consist of?** "Raw" data (e.g., unprocessed text or images) or features? In either case, please provide a description.

ScienceQA is a multi-hop QA dataset collected from elementary and high school science curricula. FOMC is a hawkish-dovish classification task, which is novel in the financial domain. The dataset is divided into three subsets: data on meeting minutes, press conference data, and speech data. We use a combination of them. MeetingBank is a new benchmark dataset for city council meeting summarization, an unstudied domain. It demands a global understanding of the whole long context. C-STANCE is the first Chinese dataset for zero-shot stance detection collected from Sina Weibo, one of the most popular Chinese social media sites. 20Minuten is a text simplification dataset consisting of full articles paired with shortened, simplified summaries from the Swiss news magazine. The corpus Py150 contains 150,000 Python programs collected from GitHub repositories. NumGLUE is an 8-task benchmark far from solved including state-of-the-art large-scale language models performing significantly worse than humans.

**Is there a label or target associated with each instance?** If so, please provide a description.

Yes.

ScienceQA: Multi choice. Labels are choice. FOMC and C-STANCE: classification. Labels are category. 20Minuten and MeetingBank: Labels are summaries. Py150: Labels are subsequence code. NumGLUE: Labels are numbers.

**Is any information missing from individual instances?** If so, please provide a description, explaining why this information is missing (e.g., because it was unavailable). This does not include intentionally removed information, but might include, e.g., redacted text.

No.

**Are relationships between individual instances made explicit (e.g., users' movie ratings, social network links)?** If so, please describe how these relationships are made explicit.

N/A

**Are there recommended data splits (e.g., training, development/validation, testing)?** If so, please provide a description of these splits, explaining the rationale behind them.

No. We split the dataset randomly.

**Are there any errors, sources of noise, or redundancies in the dataset?** If so, please provide a description.

No.

**Is the dataset self-contained, or does it link to or otherwise rely on external resources (e.g., websites, tweets, other datasets)?** If it links to or relies on external resources, a) are there guarantees that they will exist, and remain constant, over time; b) are there official archival versions of the complete dataset (i.e., including the external resources as they existed at the time the dataset was created); c) are there any restrictions (e.g., licenses, fees) associated with any of the external resources that might apply to a future user? Please provide descriptions of all external resources and any restrictions associated with them, as well as links or other access points, as appropriate.

Yes. C-STANCE is the collected from Sina Weibo and all the data are open without user privacy.

**Does the dataset contain data that might be considered confidential (e.g., data that is protected by legal privilege or by doctor-patient confidentiality, data that includes the content of individuals' non-public communications)?** If so, please provide a description.

No

**Does the dataset contain data that, if viewed directly, might be offensive, insulting, threatening, or might otherwise cause anxiety?** If so, please describe why.

No.

**Does the dataset relate to people?** If not, you may skip the remaining questions in this section.

No.

**Does the dataset identify any subpopulations (e.g., by age, gender)?** If so, please describe how these subpopulations are identified and provide a description of their respective distributions within the dataset.

N/A

**Is it possible to identify individuals (i.e., one or more natural persons), either directly or indirectly (i.e., in combination with other data) from the dataset?** If so, please describe how.

N/A

**Does the dataset contain data that might be considered sensitive in any way (e.g., data that reveals racial or ethnic origins, sexual orientations, religious beliefs, political opinions or union memberships, or locations; financial or health data; biometric or genetic data; forms of government identification, such as social security numbers; criminal history)?** If so, please provide a description.

N/A

**Any other comments?**

No additional comments.

**How was the data associated with each instance acquired?** Was the data directly observable (e.g., raw text, movie ratings), reported by subjects (e.g., survey responses), or indirectly inferred/derived from other data (e.g., part-of-speech tags, model-based guesses for age or language)? If data was reported by subjects or indirectly inferred/derived from other data, was the data validated/verified? If so, please describe how.

The TRACE benchmark comprises a series of meticulously curated tasks, specifically chosen to evaluate the continual learning capabilities of large language models across diverse domains. These tasks include domain-specific knowledge assessments, multilingual capabilities, code generation, and mathematical reasoning. Each task was selected based on its potential to challenge the models in unique ways that previous benchmarks do not address, thus filling critical gaps in the assessment of model robustness and adaptability.

To ensure uniformity and facilitate systematic evaluation, all data associated with each task instance was structured into a text-to-text format. This standardized format consists of two primary components: a query and an answer. The query component presents the problem or task to be solved by the model, ranging from complex problem statements in natural language to coding problems and mathematical queries. The answer component provides the expected response format, whether it's a direct answer, a code snippet, or a logical solution to a problem. This format not only standardizes the input-output structure for easier processing and evaluation by different models but also aligns with the common methodologies used in training and benchmarking AI systems today.

**What mechanisms or procedures were used to collect the data (e.g., hardware apparatus or sensor, manual human curation, software program, software API)?** How were these mechanisms or procedures validated?

The collection involved automated scripts to scrape and compile data from various academic and public domain sources. Manual curation was employed to ensure the quality and relevance of the tasks included in TRACE.

**What was the resource cost of collecting the data?** (e.g. what were the required computational resources, and the associated financial costs, and energy consumption - estimate the carbon footprint. See Strubell *et al.*[2] for approaches in this area.)

The resource cost associated with collecting data for the TRACE dataset was minimal, as the dataset primarily consists of samples drawn from existing datasets. The computational resources required were primarily related to the processing and preparation of these data samples, which involved standard computing equipment without the need for extensive computational power or specialized hardware.

Financial costs were similarly low, as the data leveraged publicly accessible sources, and the manipulation required was handled within the normal scope of academic research budgets. Energy consumption was limited to regular usage of academic computing resources, leading to a negligible carbon footprint for the data collection phase.

**If the dataset is a sample from a larger set, what was the sampling strategy (e.g., deterministic, probabilistic with specific sampling probabilities)?**

The TRACE dataset is a curated subset sampled from a larger collection of publicly available data sources. Given that TRACE serves as a benchmark for continual learning, the sampling strategy prioritized task challenge and diversity to effectively evaluate the models' capabilities across different domains. To ensure balance among the tasks and fairness in model evaluation, a consistent number of samples was drawn from each task category. This approach was designed to maintain uniformity in the dataset's complexity and scope, facilitating a comprehensive assessment of continual learning strategies across varied AI models.

**Who was involved in the data collection process (e.g., students, crowdworkers, contractors) and how were they compensated (e.g., how much were crowdworkers paid)?**

Data collection and curation involved contributions from graduate students and domain experts. Contributors were compensated according to academic grant stipulations or institutional pay scales, which adhere to ethical guidelines for fair compensation.

**Were any ethical review processes conducted (e.g., by an institutional review board)?** If so, please provide a description of these review processes, including the outcomes, as well as a link or other access point to any supporting documentation.

N/A.

**Does the dataset relate to people?** If not, you may skip the remainder of the questions in this section.

N/A; there is no information on people identification.

**Did you collect the data from the individuals in question directly, or obtain it via third parties or other sources (e.g., websites)?**

No.

**Were the individuals in question notified about the data collection?** If so, please describe (or show with screenshots or other information) how notice was provided, and provide a link or other access point to, or otherwise reproduce, the exact language of the notification itself.

N/A. Because TRACE is a curated subset sampled from a larger collection of publicly available data sources.

**Did the individuals in question consent to the collection and use of their data?** If so, please describe (or show with screenshots or other information) how consent was requested and provided, and provide a link or other access point to, or otherwise reproduce, the exact language to which the individuals consented.

N/A. Because TRACE is a curated subset sampled from a larger collection of publicly available data sources.

**If consent was obtained, were the consenting individuals provided with a mechanism to revoke their consent in the future or for certain uses?** If so, please provide a description, as well as a link or other access point to the mechanism (if appropriate)

N/A. Because TRACE is a curated subset sampled from a larger collection of publicly available data sources.

**Has an analysis of the potential impact of the dataset and its use on data subjects (e.g., a data protection impact analysis)been conducted?** If so, please provide a description of this analysis, including the outcomes, as well as a link or other access point to any supporting documentation.

N/A. Because TRACE is a curated subset sampled from a larger collection of publicly available data sources.

**Any other comments?**
No additional comments.

## PREPROCESSING / CLEANING / LABELING

**Was any preprocessing/cleaning/labeling of the data done(e.g.,discretization or bucketing, tokenization, part-of-speech tagging, SIFT feature extraction, removal of instances, processing of missing values)?** If so, please provide a description. If not, you may skip the remainder of the questions in this section.

Yes, we preprocess all datasets into unified format for training.

**Was the "raw" data saved in addition to the preprocessed/cleaned/labeled data (e.g., to support unanticipated future uses)?** If so, please provide a link or other access point to the "raw" data.

Yes, the original dataset can be found at:
ScienceQA:https://huggingface.co/datasets/derek-thomas/ScienceQA
FOMC: https://huggingface.co/gtfintechlab/FOMC-RoBERTa
MeetingBank:https://meetingbank.github.io/
C-STANCE:https://github.com/chenyez/C-STANCE
20Minuten:https://github.com/ZurichNLP/20Minuten
Py150:https://github.com/microsoft/CodeXGLUE
NumGLUE:https://allenai.org/data/numglue

**Is the software used to preprocess/clean/label the**

**instances available?** If so, please provide a link or other access point.

N/A

**Any other comments?**
No additional comments.

## USES

**Has the dataset been used for any tasks already?** If so, please provide a description.

Yes. All the datasets in our paper has been used for training models and evaluations performance.

**Is there a repository that links to any or all papers or systems that use the dataset?** If so, please provide a link or other access point.

N/A.

**What (other) tasks could the dataset be used for?**
The entire benchmark can be used for continual learning and our supervised tasks.

**Is there anything about the composition of the dataset or the way it was collected and preprocessed/cleaned/labeled that might impact future uses?** For example, is there anything that a future user might need to know to avoid uses that could result in unfair treatment of individuals or groups (e.g., stereotyping, quality of service issues) or other undesirable harms (e.g., financial harms, legal risks) If so, please provide a description. Is there anything a future user could do to mitigate these undesirable harms?

N/A

**Are there tasks for which the dataset should not be used?** If so, please provide a description.

N/A

**Any other comments?**
No additional comments.

## DISTRIBUTION

**Will the dataset be distributed to third parties outside of the entity (e.g., company, institution, organization) on behalf of which the dataset was created?** If so, please provide a description.

N/A

**How will the dataset will be distributed (e.g., tarball on website, API, GitHub)?** Does the dataset have a digital object identifier (DOI)?

The dataset will be distributed on Github in the future.

**When will the dataset be distributed?**

After our paper published.

**Will the dataset be distributed under a copyright or other intellectual property (IP) license, and/or under applicable terms of use (ToU)?** If so, please describe this license and/or ToU, and provide a link or other access point to, or otherwise reproduce, any relevant licensing terms or ToU, as well as any fees associated with these restrictions.
N/A

**Have any third parties imposed IP-based or other restrictions on the data associated with the instances?** If so, please describe these restrictions, and provide a link or other access point to, or otherwise reproduce, any relevant licensing terms, as well as any fees associated with these restrictions.
N/A

**Do any export controls or other regulatory restrictions apply to the dataset or to individual instances?** If so, please describe these restrictions, and provide a link or other access point to, or otherwise reproduce, any supporting documentation.
N/A

**Any other comments?**
No additional comments.

MAINTENANCE

**Who is supporting/hosting/maintaining the dataset?**
The authors and all the dataset providers.

**How can the owner/curator/manager of the dataset be contacted (e.g., email address)?**
Through email.

**Is there an erratum?** If so, please provide a link or other access point.
N/A

**Will the dataset be updated (e.g., to correct labeling errors, add new instances, delete instances)?** If so, please describe how often, by whom, and how updates will be communicated to users (e.g., mailing list, GitHub)?
Yes. We will update the dataset if there exists any errors.

**If the dataset relates to people, are there applicable limits on the retention of the data associated with the instances (e.g., were individuals in question told that their data would be retained for a fixed period of time and then deleted)?** If so, please describe these limits and explain how they will be enforced.
N/A

**Will older versions of the dataset continue to be supported/hosted/maintained?** If so, please describe how. If not, please describe how its obsolescence will be communicated to users.
Yes.

**If others want to extend/augment/build on/contribute to the dataset, is there a mechanism for them to do so?** If so, please provide a description. Will these contributions be validated/verified? If so, please describe how. If not, why not? Is there a process for communicating/distributing these contributions to other users? If so, please provide a description.
The contribution can be delivered through contacting the author or github issues https://github.com/BeyonderXX/TRACE/issues.

**Any other comments?**
No additional comments.