# OpenReview forum: "TRACE: A Comprehensive Benchmark for Continual Learning in Large Language Models"
_NeurIPS.cc/2024/Datasets_and_Benchmarks_Track — Submitted to NeurIPS 2024 Track Datasets and Benchmarks_

### Official Review · Reviewer_f8sz · 2024-07-22

**Rating:** 6
**Confidence:** 4
**Correctness:** Yes
**Clarity:** The paper is well written and easy to…

**Review:**

Strengths:

The proposed benchmark is straightforward and easy to follow. The authors conduct extensive experiments on the proposed benchmark to validate its effectiveness.

Weaknesses:

My main concern lies in the experimental results presented in the paper.

1.	The authors conduct experiments only with dense LLM model. However, the sparse MOE LLM models have achieved impressive performance across several tasks. The authors should perform experiments with sparse MOE LLMs (e.g., Mixtral, DeepSeek-V2) to comprehensively assess the continual learning abilities in LLMs.

2.	The authors should perform multi-stage continual learning across tasks and domains (general ability, instruction-following, and safety) to systematically study multi-stage continual learning abilities in LLMs.

3.	Different continual learning approaches in LLM models have varying execution times for these tasks. As for a benchmark paper, it would be interesting to see how the various LLM models affect the actual computation time. Can you provide the running time comparison among different LLM models?

**Strengths:**

The proposed benchmark is straightforward and easy to follow. The authors conduct extensive experiments on the proposed benchmark to validate its effectiveness.

**Additional Feedback:**

In Line 166, Why did you choose the LIMA dataset, and do you perform ablation studies using other alignment dataset (e.g., ChatBot Arena)?

See weaknesses in the above review.

**Documentation:**

Yes, the authors provide sufficient detail to support reproducibility.

**Limitations:**

No potential negative societal impact is detected.

**Opportunities For Improvement:**

See weaknesses in the above review.

**Relation To Prior Work:**

Yes

**Summary And Contributions:**

The authors propose a systematical benchmark for evaluating continual learning abilities in large language models (LLMs). They assess continual learning abilities in LLMs across three main domains: general ability, instruction-following, and safety. They conduct extensive experiments on eight challenge tasks and provide several highlighted takeaways.

---

> ### Author Rebuttal · Authors · 2024-08-22
>
> > W3:  Can you provide the running time comparison among different LLM models?
>
> We demonstrate the excution time of different CL methods with their training time. The results are based on Llama2-7B-Chat.
>
> | CL Method | Average Excution Time (hours) | Average Accuracy |
> | --------- | ----------------------------- | ---------------- |
> | EWC       | 129                           | 0.323            |
> | OGD       | 148                           | 0.297            |
> | PP        | 9.6                           | 0.492            |
> | L2P       | 8.4                           | 0.437            |
> | LFPT5     | 86                            | 0.492            |
> | O-Lora    | 10.1                          | 0.413            |
> | SeqFT     | 12                            | 0.487            |
> | RCL       | 12                            | **0.519**        |
>
> As shown in the table, **some regularization-based methods, such as EWC and OGD, are highly time-consuming** due to the need for storing and updating previous parameters or gradients. In contrast, O-Lora strikes a good balance between time efficiency and performance compared with EWC and OGD. Additionally, **LFPT5 can also be time-intensive due to the generation of samples from past tasks.** On the other hand, **architecture-based methods like PP and L2P are the least time-consuming**. Notably, the method proposed in this paper (RCL) demonstrates the highest overall performance.

---

> ### Author Rebuttal · Authors · 2024-08-22
>
> Thank you very much for your detailed and insightful feedback on our paper. We appreciate the time and effort you have taken to review our work. Below, we address each of your concerns in detail.
>
>
>
> > W1: The authors conduct experiments only with dense LLM model. However, the sparse MOE LLM models have achieved impressive performance across several tasks. The authors should perform experiments with sparse MOE LLMs (e.g., Mixtral, DeepSeek-V2) to comprehensively assess the continual learning abilities in LLMs.
>
> Thank you for your feedback. We conduct extra experiments on **Mixtral-8x7B-Instruct-v0.1** as your advice. The detailed results are demonstrated as follows:
>
> | Task\Round  | 1    | 2    | 3     | 4     | 5     | 6     | 7     | 8      |
> | ----------- | ---- | ---- | ----- | ----- | ----- | ----- | ----- | ------ |
> | C-STANCE    | 0.60 | 0.46 | 0.421 | 0.373 | 0.41  | 0.42  | 0.447 | 0.478  |
> | FOMC        | -    | 0.80 | 0.768 | 0.75  | 0     | 0.712 | 0.723 | 0.741  |
> | MeetingBank | -    | -    | 0.539 | 0.521 | 0.498 | 0.503 | 0.499 | 0.506  |
> | Py150       | -    | -    | -     | 0.581 | 0.579 | 0.562 | 0.546 | 0.556  |
> | ScienceQA   | -    | -    | -     | -     | 0.80  | 0.718 | 0.657 | 0.68   |
> | NumGLUE-cm  | -    | -    | -     | -     | -     | 0.48  | 0.36  | 0.39   |
> | NumGLUE-ds  | -    | -    | -     | -     | -     | -     | 0.67  | 0.61   |
> | 20Minuten   | -    | -    | -     | -     | -     | -     | -     | 0.44   |
> | **average** |      |      |       |       |       |       |       | 0.550  |
> | **BWT**     |      |      |       |       |       |       |       | -0.073 |
>
> We also demonstrate the average accuracy compared with Llama2-13B-chat and Vicuna-13B-v1.5 as follows:
>
> |                  | Average Accuracy | BWT       |
> | ---------------- | ---------------- | --------- |
> | Llama-13B        | 0.499            | **-0.07** |
> | Vicuna-13B       | 0.492            | -0.084    |
> | **Mixtral-8x7B** | **0.550**        | -0.073    |
>
> As shown in above Tabels, The average accuracy of Mixtral-8x7B is higher than Llama2-13B-chat and Vicuna-13B-v1.5 (since the model size is much larger). **However, the BWT, which refers to the degree of catastrophic forgetting, remains similar to the two models**. This demonstrates that the sparse MOE LLM models also exhibit significant catastrophic forgetting.
>
> > W2: The authors should perform multi-stage continual learning across tasks and domains (general ability, instruction-following, and safety) to systematically study multi-stage continual learning abilities in LLMs.
>
> Thank you for your constructive comments. We will present the results of a detailed multi-stage (after every task) evaluation (general ability, instruction-following, and safety) on Llama2-7B-Chat.
>
> - For general ability (We choose MMLU, GSM and MBPP)
>
> | Turn            | MMLU  | BBH   | MBPP |
> | --------------- | ----- | ----- | ---- |
> | Original        | 46.56 | 40.23 | 20.4 |
> | 1 (C-STANCE)    | 46.61 | 40.03 | 2.26 |
> | 2 (FOMC)        | 46.58 | 38.28 | 0    |
> | 3 (MeetingBank) | 46.52 | 35.34 | 0    |
> | 4 (Py150)       | 46.39 | 31.13 | 0    |
> | 5 (ScienceQA)   | 46.44 | 39.46 | 0    |
> | 6 (NumGLUE-cm)  | 46.31 | 38.02 | 0    |
> | 7 (NumGLUE-ds)  | 46.53 | 34.13 | 0    |
> | 8 (20Minuten)   | 46.43 | 30.11 | 0    |
>
> As shown in the table, the factual knowledge (MMLU) remains stable during the training. The performance of general reasoning (BBH) gradually decreases while increasing after training on ScienceQA, which is the motivation of our RCL method. In addition, the code generation (MBPP) capability quickly declines to 0.
>
> - For instruction-following, we report the win, tie, lose rate against models without training.
>
> | Turn            | Win  | Tie  | Lose |
> | --------------- | ---- | ---- | ---- |
> | Original        | 0%   | 100% | 0%   |
> | 1 (C-STANCE)    | 19%  | 60%  | 21%  |
> | 2 (FOMC)        | 15%  | 59%  | 26%  |
> | 3 (MeetingBank) | 16%  | 48%  | 36%  |
> | 4 (Py150)       | 12%  | 44%  | 44%  |
> | 5 (ScienceQA)   | 12%  | 46%  | 42%  |
> | 6 (NumGLUE-cm)  | 15%  | 37%  | 48%  |
> | 7 (NumGLUE-ds)  | 14%  | 31%  | 55%  |
> | 8 (20Minuten)   | 14%  | 34%  | 53%  |
>
> As shown in the Table, the instruction-following gradually decrease during the training, with lose rate increase step-by-step.
>
> - For safety, we report the win, tie, lose rate against models without training.
>
>   | Turn            | Win  | Tie  | Lose |
>   | --------------- | ---- | ---- | ---- |
>   | Original        | 0%   | 100% | 0%   |
>   | 1 (C-STANCE)    | 1%   | 97%  | 2%   |
>   | 2 (FOMC)        | 0%   | 98%  | 2%   |
>   | 3 (MeetingBank) | 0%   | 99%  | 1%   |
>   | 4 (Py150)       | 0%   | 100% | 0%   |
>   | 5 (ScienceQA)   | 1%   | 99%  | 0%   |
>   | 6 (NumGLUE-cm)  | 0%   | 96%  | 4%   |
>   | 7 (NumGLUE-ds)  | 0%   | 97%  | 3%   |
>   | 8 (20Minuten)   | 0%   | 98%  | 2%   |
>
> As shown in the Table, training on our benchmark (without harmful dataset) has a marginal influence on the model safety.

---

> ### Author Response · Authors · 2024-08-27
> **Follow-up on Rebuttal Response**
>
> Dear Reviewer f8sz,
>
> I understand that the review process is time-consuming and appreciate the effort you are putting into evaluating my work.
>
> Since the rebuttal period is nearing its end, I wanted to kindly check if there are any further clarifications or additional information you might need from my side to assist with the review.
>
> Your feedback is highly valuable, and I am eager to address any remaining concerns you may have.
>
> Best regards,
>
> Authors

---

> ### Comment · Reviewer_f8sz · 2024-08-28
> **Response to Authors**
>
> Thanks for the authors' response. I have no further questions. Based on the current version of this paper, I will raise my evaluation score. Additionally, I strongly recommend that the authors include all these experimental findings in the main text of the paper. This would offer a more comprehensive assessment of LLM models and bolster the overall argument of the paper.

---

> > ### Author Response · Authors · 2024-08-30
> > **Response to Reviewer f8sz**
> >
> > Dear Reviewer f8sz,
> >
> > Thank you very much for your thoughtful response and positive feedback. We will certainly incorporate these experimental findings into the revised version of our paper to provide a more comprehensive evaluations.
> >
> > We sincerely appreciate your constructive comments. Your support and guidance have been invaluable to us.
> >
> > Sincerely,
> >
> > Authors.

---

### Official Review · Reviewer_Rk5m · 2024-07-23
**Benchmark for Continual Learning capabilities in LLM**

**Rating:** 7
**Confidence:** 3
**Correctness:** Yes
**Clarity:** Yes

**Review:**

The paper is well-written and well-organized, and it brings up an essential problem for LLMs. The paper extensively evaluates different methods on the benchmark dataset and presents exciting conclusions.

Questions.
* The description for the proposed metrics (e.g., 'General Ability Delta') is unclear. I think it would be beneficial for the readers first to understand the metrics and the intuition behind them.

* Some results in the paper are not very clear. For example, the OP of LoRA in Table 2 is low. Does it mean that the LoRA does not have the capability to learn the current task? If so, then maybe better hyperparameters might resolve the issue and give a fair comparison to LoRA.

* In Table 2, some CL algorithms fail to achieve the performance of SeqFT; could you explain the reason behind it?

**Strengths:**

Explained in the previous sections

**Additional Feedback:**

NA

**Documentation:**

Yes

**Limitations:**

Yes

**Opportunities For Improvement:**

I believe a better explanation of CL baselines, metrics, and tables can improve the paper.

**Relation To Prior Work:**

Yes

**Summary And Contributions:**

The paper introduces a new benchmark dataset to evaluate the effect of continual training on LLMs. It is already known that neural networks are prone to catastrophic forgetting. However, only a few works have explored this phenomenon for LLMs. The paper argues that existing benchmarks are not comprehensive and challenging enough; therefore, a new benchmark is required.

* The benchmark has some characteristics that distinguish it from the previous ones. It covers more complex and critical tasks, including domain-specific tasks, multilingual capabilities, code generation, and mathematical reasoning.

* It covers a wide range of existing CL and training methods as its baseline, which helps understand the extent of their conclusions.

* The authors present their include a wide veriety of available LLMs.

---

> ### Author Rebuttal · Authors · 2024-08-22
>
> We appreciate your feedback and have carefully considered your suggestions.
>
> > W1: The description for the proposed metrics (e.g., 'General Ability Delta') is unclear. I think it would be beneficial for the readers first to understand the metrics and the intuition behind them.
>
> Thank you for your valuable suggestion. We will address this in the next version of our paper by providing a more detailed explanation of the metrics.
>
> The 'General Ability Delta' describes the change in general abilities of the LLM, such as Factual Knowledge and Commonsense Reasoning, before and after continual training.
>
> > W2: Some results in the paper are not very clear. For example, the OP of LoRA in Table 2 is low. Does it mean that the LoRA does not have the capability to learn the current task? If so, then maybe better hyperparameters might resolve the issue and give a fair comparison to LoRA.
>
> We appreciate your insightful feedback. To address this, we conducted additional experiments on LLaMA2-7B by increasing the LoRA rank dimension from 8 to 16. The results are as follows:
>
> |                      | OP | BWT    |
> | -------------------- | ---------------- | ------ |
> | LoRA (d=8, lr=1e-4)  | 0.127            | -0.457 |
> | LoRA (d=12, lr=1e-4) | 0.272            | -0.296 |
> | LoRA (d=16, lr=1e-4) | **0.312**        | -0.274 |
>
> The table above indicates that increasing the LoRA rank dimension helps mitigate catastrophic forgetting, suggesting that severe forgetting in LoRA may be due to the small number of trainable parameters. This reduction in forgetting could be attributed to the increase in trainable parameters, which reduces the overlap of neuron updates corresponding to different tasks.
>
> > W3: In Table 2, some CL algorithms fail to achieve the performance of SeqFT; could you explain the reason behind it?
>
> The suboptimal performance of traditional continual learning methods such as EWC and OGD might be due to their reliance on LoRA. As explained on line 171, “EWC and OGD require storing parameters or gradients from past tasks, which is impractical for the full parameter setting of LLMs. Therefore, we validate these methods using LoRA.”

---

> > ### Comment · Reviewer_Rk5m · 2024-08-31
> > **Acknowledge the rebuttal**
> >
> > Thank the authors for their efforts during the rebuttal process. I will keep my rating as **7**.
> >
> > This paper is well-written, and the benchmark can help the LLM community focus on more realistic problems and challenges of continual learning in LLM.

---

### Official Review · Reviewer_8LT7 · 2024-07-25
**A reasonable dataset and analysis, but the problem with catastrophic forgetting of LLMs on continual learning seems to be known**

**Rating:** 5
**Confidence:** 4

**Review:**

The pros
- catastrophic forgetting of LLMs is a serious issue and must be taken into consideration in several scenarios
- paper does an evaluation of the general, instruction-following and safety abilities of LLMs in the continual learning scenario
- a reasoning-augmented CL method is proposed and briefly evaluated.

The cons
- it is not clear how original is this work, since there are surveys on continual learning of LLMs, which seem to highlight this issue already
- it is not clear how (or whether) the authors ensure the desiderata for the eight chosen datasets.

The paper is understandable with some effort. The authors define a few characteristics for choosing the datasets in TRACE. [lines 100-102: First, the datasets should be novel enough that most LLMs have not been trained on them. Second, they should be challenging for large language models. Third, a variety of tasks should be covered in our benchmark.]

However, they have not adequately justified whether the chosen datasets satisfy these criteria. For example, how did the authors ensure that the first and second criteria were satisfied for the combination of datasets and LLMs. The authors presumably feel that eight is a good enough number to cover the variety criterion.

The main issues in the presentation are that there are some undefined acronyms (e.g., EM, GP, etc.) and some inconsistencies (e.g., the text states 0-shot F1 for TyDiQA in line 178, whereas the Table 3 has 1-shot F1 for the TyDiQA column).

The work is clearly significant for all those fine tuning all or some of the parameters of LLMs and expecting the original performance to stay intact. However, the originality of this work is unclear, since there are some recent surveys on the topic of Continual Learning for LLMs in 2023 and early 2024, which seem to mention the catastrophic forgetting issue for LLMs (e.g., the survey of Tongtong Wu et al. in Arxiv on 7 Feb 2024). Of course, catastrophic forgetting for continual learning of neural networks is well known.

**Strengths:**

The tasks and datasets chosen in TRACE could be a nice strength of this paper, if they are properly justified. The metrics defined are quite straightforward.

**Additional Feedback:**

The choice of LLMs could also be better justified.

**Clarity:**

The paper is understandable with some effort. There is a long appendix with more details.

**Correctness:**

The paper seems sound for the most part, although there are a few undefined concepts and inconsistencies.

**Documentation:**

The documentation could be improved,.

**Ethics:**

Minimal concerns.

**Limitations:**

Limitations section (Section 6) is included by the authors, which mentions potential biases and a couple of other issues.

**Opportunities For Improvement:**

As mentioned above in my review, the authors should strengthen the justification of the chosen datasets and the originality claims.

**Relation To Prior Work:**

This area seems to be a bit problematic since the survey on the topic of the paper that was cited in the paper is a bit dated (2022 - reference 22 of the paper).

**Summary And Contributions:**

The authors create TRACE, a collection of eight datasets that includes a set of tasks for continual learning, covering
challenges in domain-specific tasks, multilingual capabilities, code generation, and mathematical
reasoning. They also perform a post-training evaluation of LLM capabilities through other datasets, which include Factual Knowledge:
using MMLU dataset [40], General Reasoning: evaluated with BBH [41], Multilinguality:
 using TyDiQA [42], a multilingual QA benchmark across 11 languages, Commonsense Reasoning: assessed with PIQA [43], Code Generation: using MBPP [44],  Reading Comprehension: using BoolQ [45], In addition to traditional continual
learning metrics, they introduce General Ability Delta, Instruction Following Delta, and Safety Delta
to evaluate shifts in LLM’s inherent abilities. A reasoning-augmented Continual learning (CL) method is also proposed in Section 4.6 and then briefly evaluated.

---

> ### Author Rebuttal · Authors · 2024-08-22
>
> Thank you for your appreciation and an excellent summarization of our work. We appreciate your time and effort in carefully reviewing our paper. We address your remarks below:
>
> > W1: it is not clear how original is this work, since there are surveys on continual learning of LLMs, which seem to highlight this issue already.
>
> Our work **differs from existing continual learning surveys** by specifically proposing a benchmark for **continual instruction tuning** and conducting extensive evaluations. While surveys summarize various settings like continual pre-training and continual instruction tuning, as well as the challenges and solutions therein, they **do not construct datasets tailored for continual learning of LLMs.**
>
> As stated in line 32 of our paper, previous NLP continual learning benchmarks, such as AG News, Amazon Reviews, and Yelp Reviews, present limited difficulty for contemporary LLMs and are primarily NLU tasks. Moreover, due to their antiquity, these datasets pose a significant risk of data leakage and are thus unsuitable as benchmarks for LLM continual learning. **TRACE, on the other hand, emphasizes novelty, challenge, and task diversity in dataset selection.**
>
> Additionally, should it be necessary, the publication date of our preprint can substantiate the novelty of our work.
>
> > W2: it is not clear how (or whether) the authors ensure the desiderata for the eight chosen datasets. They have not adequately justified whether the chosen datasets satisfy these criteria.
>
> This is a good question! **Ensuring the challenge and avoiding data contamination are crucial.** The models we evaluated, such as LLaMA2, Baichuan 2, and Vicuna-V1.5, were all released in 2023, and our chosen task datasets primarily from 2023 and 2022. Although we cannot ascertain the exact training data used by these LLMs or rule out possible data contamination, we verified based on the final performance of the models.
>
> Before finalizing our dataset choices, we conducted extensive validation, retaining only those datasets where these models underperformed, indicating their challenge and likely lack of prior exposure.
>
> **From Tables 2 and 6, we observe that the Overall Performance (OP) of these models is around 40, underscoring the datasets' difficulty and suggesting minimal prior training on this data.**
>
> > W3: The main issues in the presentation are that there are some undefined acronyms (e.g., EM, GP, etc.) and some inconsistencies (e.g., the text states 0-shot F1 for TyDiQA in line 178, whereas Table 3 has 1-shot F1 for the TyDiQA column).
>
> Thank you for your valuable feedback. We will correct these errors and ensure thorough verification in subsequent versions.

---

> ### Author Response · Authors · 2024-08-27
> **Follow-up on Rebuttal Response**
>
> Dear Reviewer 8LT7,
>
> I understand that the review process is time-consuming and appreciate the effort you are putting into evaluating my work.
>
> Since the rebuttal period is nearing its end, I wanted to kindly check if there are any further clarifications or additional information you might need from my side to assist with the review.
>
> Your feedback is highly valuable, and I am eager to address any remaining concerns you may have.
>
> Best regards,
>
> Authors

---

> ### Author Response · Authors · 2024-08-30
> **Looking forward to your feedback!**
>
> Dear reviewer 8LT7,
>
> We greatly appreciate your feedback on the paper and hope to have addressed your questions and comments in the response. We would like to kindly remind you to let us know if you have any further questions or concerns after reading the responses. We will be happy to answer them.
>
> Thank you again for your time.
>
> Authors

---

### Official Review · Reviewer_eEUp · 2024-08-02

**Rating:** 6
**Confidence:** 3
**Correctness:** yes
**Clarity:** yes

**Review:**

Pros:
1. The main contribution is the constructed dataset. I agree with the statement of the author that, the main contribution of the dataset comes from three perspectives: 1. models should have not been trained on them. 2. the tasks should be challenging 3. the tasks should be diverse.
2. The observation on the impact of reasoning and the proposal of RCL is somehow novel


Cons:
The conclusion here is somehow less supervising. Like in general the catastrophic forgetting still exists, or replay is the best.

**Strengths:**

see reviews

**Additional Feedback:**

Question:
1. For replay-based sequential finetuning, why it is not further divided into full finetune and lora finetune?

**Documentation:**

NA

**Limitations:**

yes

**Opportunities For Improvement:**

This can be a great tools but at this point it is hard for me to learn much from this benchmark....

**Relation To Prior Work:**

NA

**Summary And Contributions:**

The authors introduce a benchmark TRACE to assess the continual learning capabilities of existing LLMs. The dataset designed to evaluate models is their main contribution, which consists of tasks from multilingual capabilities, code generation, mathematical reasoning and domain specific tasks. Their evaluation includes both in-distribution test and generalization test, which concludes that most existing LLMs stills struggles in achieving a good continual learning ability, but tasks augmented with reasoning path are relatively affective. Inspired by this observation, they propose reasoning-augmented continual learning which achieves significant boost.

---

> ### Author Rebuttal · Authors · 2024-08-22
>
> Thank you for your constructive feedback. We try to address your remarks below:
>
> > W1: The conclusion here is somehow less supervising. Like in general the catastrophic forgetting still exists, or replay is the best.
>
>  We acknowledge that some of our experimental results might appear less supervising. However, we believe that **our study offers empirical validation of continual learning performance in LLMs**, showing that their behavior regarding catastrophic forgetting is similar to that of smaller models.
>
> Beyond the observation that "catastrophic forgetting still exists, or replay is the best," our research highlights several interesting findings:
>
> 1. **Cross-lingual Transfer Ability** : As highlighted in Table 3, the performance on TydiQA, a dataset spanning 11 languages, interestingly improved, contrasting with results from other datasets. Despite our training data including C-STANCE (Chinese) and 20Minuten (German), we observed performance gains across other languages in TydiQA. This suggests a notable cross-lingual transfer ability of LLMs even with limited language data. We have emphasized this phenomenon on line 225.
> 2. **Evaluation of Traditional Continual Learning Methods** : Our findings in Table 2 demonstrate that **some traditional continual learning methods do not work well in the context of LLMs** . Techniques like EWC and OGD showed no significant advantages over the baseline, LoRASeqFT, which does not employ any forgetting mitigation measures.
> 3. **Safety Assessment** :  In the Safety evaluation presented in Figure 2, we observed that **the safety of models trained on sequential tasks did not deteriorate.** This is a significant conclusion, demonstrating that as long as LLMs are not fine-tuned on hazardous datasets, their safety remains unchanged. In other words, **during the fine-tuning process, the models learn task-specific information without adopting a behavior pattern of responding to every query indiscriminately** . For companies like OpenAI, this finding suggests that by ensuring the filtration of dangerous user-uploaded data, the risk of models producing harmful responses post fine-tuning is minimized.
> 4. **Reasoning Paths for Continual Learning** : As you noted in your pros, the effectiveness of reasoning paths in enhancing continual learning is a valuable insight for future LLM continual learning strategies.

---

> > ### Comment · Reviewer_eEUp · 2024-08-28
> >
> > Thanks for your clarification. I will raise my score and I suggests put all those findings in a more obvious way.

---

> > > ### Author Response · Authors · 2024-08-30
> > > **Response to Reviewer eEUp**
> > >
> > > Dear Reviewer eEUp,
> > >
> > > Thank you very much for your thoughtful response and positive feedback. We will certainly incorporate these findings into the revised version of our paper to ensure much more contributions of our work.
> > >
> > > We sincerely appreciate your constructive comments. Your support and guidance have been invaluable to us.
> > >
> > > Sincerely,
> > >
> > > Authors.

---

> ### Author Response · Authors · 2024-08-27
> **Follow-up on Rebuttal Response**
>
> Dear Reviewer eEUp,
>
> I understand that the review process is time-consuming and appreciate the effort you are putting into evaluating my work.
>
> Since the rebuttal period is nearing its end, I wanted to kindly check if there are any further clarifications or additional information you might need from my side to assist with the review.
>
> Your feedback is highly valuable, and I am eager to address any remaining concerns you may have.
>
> Best regards,
>
> Authors

---

### Decision · Program_Chairs · 2024-09-26

**Decision:**

Reject

**Comment:**

The paper introduces a new benchmark dataset to evaluate the effect of continual training on LLMs. This is done by providing some additional relatively standard finetuning dataset and showing how finetuning on those might degrade preformance of the LLMs on the other common evaluation metrics. It is giving both benchmark results as well as some new methodologies (where I'd prefer to separate these in this track though). I would have hoped for a bit more discussion on how much of the effect of 'forgetting' is tied to special output formats of the tasks (such as MCQ answers), or the lengths of each answer (not the question) in the dataset.

Also, the authors need to improve the related work discussion to more recent works. This would also benefit the main contribution of an up-to-date benchmarking here in this case. Please also comment on how the datasets will be made available for the community to keep the benchmark alive.

Given these issues, I cannot recommend acceptance.